# Nanoparticle- and Microparticle-Based Vaccines against Orbiviruses of Veterinary Importance

**DOI:** 10.3390/vaccines10071124

**Published:** 2022-07-14

**Authors:** Luis Jiménez-Cabello, Sergio Utrilla-Trigo, Natalia Barreiro-Piñeiro, Tomás Pose-Boirazian, José Martínez-Costas, Alejandro Marín-López, Javier Ortego

**Affiliations:** 1Centro de Investigación en Sanidad Animal (CISA-INIA/CSIC), 28130 Madrid, Spain; luisfjim@ucm.es (L.J.-C.); sergio.utrilla@inia.csic.es (S.U.-T.); 2Centro Singular de Investigación en Química Biológica y Materiales Moleculares (CIQUS), Universidad de Santiago de Compostela, 15782 Santiago de Compostela, Spain; natalia.barreiro.pineiro@usc.es (N.B.-P.); tomas.pose@rai.usc.es (T.P.-B.); jose.martinez.costas@usc.es (J.M.-C.); 3Section of Infectious Diseases, Department of Internal Medicine, Yale University School of Medicine, New Haven, CT 06519, USA; alejandro.marinlopez@yale.edu

**Keywords:** reovirus, orbivirus, bluetongue, African horse sickness, epizootic hemorrhagic disease, subunit vaccine, nanocarrier, microcarrier

## Abstract

Bluetongue virus (BTV) and African horse sickness virus (AHSV) are widespread arboviruses that cause important economic losses in the livestock and equine industries, respectively. In addition to these, another arthropod-transmitted orbivirus known as epizootic hemorrhagic disease virus (EHDV) entails a major threat as there is a conducive landscape that nurtures its emergence in non-endemic countries. To date, only vaccinations with live attenuated or inactivated vaccines permit the control of these three viral diseases, although important drawbacks, e.g., low safety profile and effectiveness, and lack of DIVA (differentiation of infected from vaccinated animals) properties, constrain their usage as prophylactic measures. Moreover, a substantial number of serotypes of BTV, AHSV and EHDV have been described, with poor induction of cross-protective immune responses among serotypes. In the context of next-generation vaccine development, antigen delivery systems based on nano- or microparticles have gathered significant attention during the last few decades. A diversity of technologies, such as virus-like particles or self-assembled protein complexes, have been implemented for vaccine design against these viruses. In this work, we offer a comprehensive review of the nano- and microparticulated vaccine candidates against these three relevant orbiviruses. Additionally, we also review an innovative technology for antigen delivery based on the avian reovirus nonstructural protein muNS and we explore the prospective functionality of the nonstructural protein NS1 nanotubules as a BTV-based delivery platform.

## 1. Introduction

The order *Reovirales* is a heterogeneous non-enveloped virus group that comprises a variety of viruses widely distributed in nature through a diversity of hosts, ranging from birds, fish and mammals to insects, fungi and plants, leading to different pathological manifestations. Viruses in this family share several common characteristics, such as replication in the cytoplasm of infected cells using similar strategies for both genome replication and expression of viral proteins, possession of a fragmented genome composed of 9 to 12 double-stranded RNA fragments (dsRNA), and the presence of two or three concentrically arranged protein shells with an external diameter of 60–90 nm [1,2,3]. To date, based on the structural characteristics of the virions, the number of genomic segments and the strategies of the replicative cycle, the members of this order have been grouped into two families that comprise fifteen different genera: *Spinareoviridae* (*Orthoreovirus*, *Cypovirus*, *Oryzavirus*, *Idnerovirus*, *Aquareovirus*, *Coltivirus*, *Fijivirus*, *Dinovernavirus* and *Mycoreovirus*) and *Sedoreoviridae* (*Orbivirus*, *Rotavirus*, *Phytoreovirus*, *Cardoreovirus*, *Seadornavirus* and *Mimoreovirus*). Only viruses of the *Orthoreovirus*, *Orbivirus*, *Rotavirus*, *Seadornaviruses* and *Coltivirus* genera are capable of infecting humans and other vertebrates [2,4,5,6], whereas the remaining genera infect the rest of the aforementioned hosts.

Within the veterinary field, viruses belonging to the genera *Orthoreovirus*, *Orbivirus* and *Rotavirus* stand out for their noteworthy impact. The genus *Orbivirus* is composed of viruses that possess ten genomic dsRNA segments and present a structure characterized by its icosahedral capsid (~90 nm in diameter), which is divided into three concentric layers [7,8]. The outer capsid of the virus is formed by VP2, the most exposed virion protein, involved in virus entry and the main determinant of serotype specificity, along with VP5, also involved in virus entry [9,10]. The inner capsid or core is composed of two additional layers, the subcore formed by VP3 and the intermediate layer constituted by VP7, along with three minor structural proteins with enzymatic activities, VP1, VP4 and VP6 [8,11,12,13]. Several members of the *Orbivirus* genus cause severe disease in mammals. However, three Culicoides-transmitted arboviruses have gathered more attention due to their significantly high economic impact: bluetongue virus (BTV), African horse sickness virus (AHSV) and epizootic hemorrhagic disease virus (EHDV).

In light of the absence of therapeutic treatments [14,15,16], vaccination is the most effective prophylactic measure against these three relevant ruminant diseases [17,18,19,20]. Classic vaccine approaches (live attenuated and inactivated vaccines) have contributed to the control of these viral diseases. However, important safety pitfalls and limited protection throughout the diversity of serotypes of each virus constrain their use and push forward the development of next-generation vaccines. During the last few decades, a wide range of vaccine candidates have been designed based on a variety of technologies. Recently, novel approaches have emerged by the genetic modification of orbiviruses. The development of reverse genetics systems allowed the production of disabled infectious single cycle (DISC) and disabled infectious single animal (DISA) vaccines [18,19], showing promising results in BTV and AHSV natural hosts [18]. Viral vector and subunit vaccines have also been shown to be efficacious in natural hosts [17,18]. Baculovirus and plant-based expression systems have been implemented for a generation of diverse nanoparticle and microparticle-based vaccines against BTV, AHSV and EHDV.Here, we review the nanoparticle- or microparticle-based vaccine candidates developed so far against these three highly veterinary impact orbiviruses, as well as potential delivery platforms that have emerged from the study of these or other *Reoviridae* members.

## 2. Nanoparticle- and Microparticle-Based Vaccine Platforms

Antigen delivery by micro- or nanoparticles is a versatile cutting-edge alternative for prompting responses to emerging or re-emerging pathogens, disease outbreaks, epidemics and pandemics, as they are safe (absence of infectious material), DIVA (differentiating between infected and vaccinated animals), are thermally and chemically stable and large-scale production is affordable. A variety of technologies can be used as antigen carriers or can be easily conjugated with epitopes to tune their chemistry and immunogenicity. Antigen-carrying lipid particles include liposome-based delivery systems that, aside from their usage as nucleic acid carriers [21,22,23] or upgraders of adjuvant efficacy [24,25], have been widely explored as promising tools for vaccine design [26,27,28,29]. Extracellular vesicles (EVs), small lipid-based bilayer particles that are naturally secreted by all cell types, virosomes and lipid squalene nanoparticles also represent potential lipid-based delivery systems [30,31,32]. Non-lipidic biodegradable nanocarrier- or microcarrier-based delivery platforms have been also assessed, mainly carbohydrate constructs and dendrimers. The most common polymeric nano- and microparticles are those based on poly(α-hydroxy acids) such as the Food and Drug Administration (FDA) and European Medicine Agency (EMA)-approved poly(lactic-coglycolic acid) (PLGA) or poly(lactic acid) (PLA)-based nanoparticles [33,34,35,36,37,38,39]. In addition, poly-amino acids, such as poly(γ-glutamic acid) (γ-PGA) [40], poly(ε-lysine) [41] or poly(L-arginine) [42], and polysaccharides such as chitosan, alginate or dextran [43,44,45,46,47], have also been developed. Dendrimers (usually olypropyleneimine (PPI) and polyamido amine (PAMAM) dendrimers) are also attractive platforms due to their well-defined branched structure and structural versatility that offer the possibility of encapsulating or coupling functional vaccine antigens [48,49,50,51,52]. Additionally, inorganic compounds such as gold (AuNPs), silver (AgNPs) [53,54,55], silica [56] or carbon [57] can be used for the safe, reproducible and low-cost generation of nanosized structures for enhanced antigen delivery [52].

Self-assembled protein-based nanoparticle systems include several approaches such as viral-like particles (VLPs), heat shock protein “cages” [58], enzyme complexes [59], chaperones and the highly conserved ferritin, which forms stable tetracosameric protein shells which can include heterologous antigens [60,61]. Nonetheless, VLPs have gathered the most attention, with more than 110 VLPs constructed from viruses belonging to a wide diversity of families [62]. VLPs are nanoscale structures made up of self-assembled viral proteins that lack the genetic material, being replication-deficient and non-pathogenic in consequence. VLPs present a well-defined geometry, usually icosahedrons or rod-like structures with diameters in the range of 25–100 nm [63], mimicking the overall organization and conformation structure of the native virions, thereby resembling the immunogenic features of the natural virus [64]. These protein structures are based on the natural intrinsic ability of several structural viral proteins, frequently major proteins in the capsid or envelop, and also core proteins which spontaneously self-assemble into VLPs when they are expressed in recombinant expression systems [65,66]. VLPs can be used as vaccine candidates or vectors for the presentation of foreign antigens or ligands or for drug delivery [67]. VLPs represent pathogen-associated molecular patterns (PAMPs) due to their repetitive and multimeric structures that resemble wild-type viruses, being detected by the host, directly interacting with antigen-presenting cells (APCs), as dendritic cells (DCs), and triggering an adaptive immune response. In addition, they present a higher stability than soluble antigens and have DIVA properties, as VLPs do not incorporate all the viral antigens. Due to these features, VLPs can be exploited as safe and versatile vaccine candidates against viral infections. Importantly, VLPs imply some significant technical challenges, e.g., low stability and instability throughout the downstream processing, variable expression of different proteins between expression systems and co-purification of impurities that may affect VLP immunogenicity [68]. Since 1986, when the FDA first approved a recombinant VLP-based vaccine against hepatitis B virus (HBV), several VLPs against different infectious diseases followed suit: Gardasil^®^ (Merck and Co., Inc., Kenilworth, NJ, USA), Gardasil9^®^ (Merck and Co., Inc.) and Cervarix^®^ (GlaxoSmithKline Inc., Brentford, UK) against human papilloma virus; Sci-B-Vac™ (VBI Vaccines, Cambridge, MA, USA) and Hecolin^®^ (Xiamen Innovax Biotec, Xiamen, China) against Hepatitis B and E viruses, respectively; Mosquirix™ (Glaxo-SmithKline Inc.) against Plasmodium falciparum, the malaria agent; and many others currently being evaluated in clinical trials (reviewed in [67]). VLPs are increasingly being considered as veterinary vaccines. To date, only porcine circovirus type 2 (PCV2) VLP-based vaccines Porcilis PCV^®^ (Intervet International, Boxmeer, The Netherlands) and Ingelvac CircoFLEX1^®^ (Boehringer Ingelheim, Ingelheim am Rhein, Germany) have been licensed for veterinary purposes [69]. However, VLPs have been generated from a broad spectrum of animal viral diseases, such as AHSV [70], avian influenza virus (AIV) [71], BTV [72], foot and mouth disease virus (FMDV) [73], Newcastle disease virus (NDV) [74], porcine parvovirus (PPV) [75] and Rift Valley fever virus (RVFV) [76], among others reviewed in [77,78].

Self-assembled protein nano- or microplatforms can be experimentally generated under laboratory conditions in a range of cell-based expression systems including mammalian cell lines, prokaryotic cells, yeast and insect cell lines [79]. Bacteria was the first expression platform used for the expression of biologics—pharmaceutical compounds synthesized or extracted from a biological source, e.g., monoclonal antibodies, receptor modulators, enzyme modulators or vaccines [80]—and is the most used due to their rapid growth, ease to handle and low cost [79,81,82]. However, they are unable to perform post-translational modifications, there is a potential of contamination with endotoxins and the “rare” codon usage or incorrect protein folding compared to eukaryotic cells may lead to low protein levels or useless products [83]. In yeast, misfolded protein has been related with high cell density culture [84]. Mammalian and insect cells lines achieve high protein yields and can perform post-translational modifications, although in some mammalian cells and in insect cells lines these modifications can differ from those of human cells, leading to immunogenic biologics that can be recognized as “foreign” by the immune system [79,85]. Molecular farming—a term used to describe the use of plants or plant cells to produce biologics [86]—has been implemented for the production of VLPs-based vaccines against a plethora of viruses (reviewed in [87,88]). Transient expression in plants permitted the successful expression of naked and enveloped particles, and structures depending on one or more viral proteins [88]. The advantages of this technology, e.g., its safety compared to the aforementioned expression systems, the possibility of engineering human-type glycosylation and its easy scalability [89], make it more attractive and accessible for manufacturing self-assembled protein nano- or microparticle vaccines. Recently, the first large-scale studies of any plant-derived human vaccine—quadrivalent VLP against influenza virus—showed that a plant-derived VLP can provide substantial protection against a viral pathogen without safety concerns [90]. Other VLP vaccine candidates against rotavirus and SARS-CoV-2 are under phase I, II and III clinical evaluation [91,92].

### µNS Avian Reovirus Protein as an Antigen Carrier System

One of the genera that compose the family *Spinareoviridae* is the genus *Orthoreovirus*. Viruses belonging to this genus possess ten genomic dsRNA segments inside a double protein coat of 70–90 nm in external diameter [93]. The main non-fusogenic mammalian reoviruses (MRVs) and fusogenic avian reoviruses (ARVs) are the prototype members of this genus. ARVs, like all other reoviruses, create membrane-less intracellular compartments for viral replication, which are named “viral inclusion bodies” or “viroplasms” [94]. The scaffold upon which viroplasms are constructed is created by the viral non-structural protein muNS, which first attracts the RNA-binding, non-structural protein sigmaNS, and then together they selectively and orderly load the viroplasms with all other viral components [95,96]. Protein muNS is encoded by the single open reading frame of ARV gene M3 [95]. This 635 amino acid non-structural viral protein (70.8 kDa) is capable of forming ordered inclusions when expressed individually in transfected cells and it shares only a 28.3% similarity with its mammalian reovirus counterpart. The C-terminal region of the muNS sequence is essential for the formation of inclusions [97]. Conversely, no change is observed in the capacity to form inclusions or the shape and behavior of those after the elimination of the N-terminal first 140 amino acids. However, further elimination of N-terminal residues up to residue 448 resulted in the formation of smaller, regular and spherical inclusions. Any additional deletion resulted in a total loss of the ability to form inclusions, thus showing that the fragment between residues 448 and 635 is the minimal portion of muNS protein that remains capable of forming inclusions efficiently and it was subsequently named muNS-Mi [97]. MuNS-Mi were named muNS-microspheres (muNS-MS) due to their size being in the micrometer range when they are produced in the baculovirus–insect cell expression system (Figure 1). MuNS-Mi sequence analysis revealed the presence of two different regions with a high probability of forming coiled-coil structures that were designated as domains Coil1 (C1) and Coil2 (C2). The domain in between C1 and C2 was called Intercoil domain or IC (residues 477 to 539), whereas the C-terminal 61 residues comprised domain C-Tail (CT). The IC domain is one of the main actors in the inclusion formation, as point mutations in two highly conserved residues (His 487 and Cys 489) led to the complete loss of the ability to form inclusions [97]. As the IC domain showed very high affinity for muNS, muNS-Mi inclusions and muNS-MS, it was subsequently used as a molecular “tag” [98,99]. The so-called “IC-Tagging” methodology is made up of two components: the protein muNS-Mi that forms intracellular muNS-MS when expressed alone; and the IC domain that can be fused to any protein of interest at either the amino or carboxyl terminus without altering the protein folding, activity or its cellular location. The IC tag forces the relocation of the tagged protein to the muNS-MS when co-expression of these two components occurs. Proteins included into the muNS-MS by IC-Tagging fold properly and even oligomer formation occurs.

Although the platform was originally developed in eukaryotic cells, where diameters of the muNS-MS range between 1 and 4 μm, it was also adapted to work in bacteria, where the spheres have a diameter of around 400 nm (nanospheres, muNS-NS) (Figure 1). Therefore, this technology could also benefit from the capabilities of nanosize-particles to optimize antigen uptake or prompt Th1 responses. In addition, the bacterial version is easy to handle and cost-effective for protein production, adding versatility to the methodology [100], and allows for the production of encapsulated proteins or enzymes for different uses [101]. Additionally, the IC-Tagging methodology was adapted to work inside the endoplasmic reticulum (ER). By using the G glycoprotein of vesicular stomatitis virus (VSV) and protein Gn from Rift Valley fever virus (RVFV), it was demonstrated that fully glycosylated proteins can be loaded in ER muNS-MS, thus mimicking the surface of enveloped viruses [100].

The IC-Tagging methodology also presents a number of competitive advantages—such as its simplicity, cost effectiveness and easy production of insoluble and difficult-to-express proteins—for the production of proteins immobilized in particulate form in relation to other techniques, such as synthetic nanoparticles or VLPs. Moreover, these particles lack structural constraints. Some of the existing methods, e.g., VLPs, imply architectural limits about how much foreign protein is admitted and their load, and even might influence the proper folding of the target peptide. The muNS-MS and muNS-NS are ordered aggregates with no particular structural restrictions that admit quaternary interactions between monomers, proteins with enzymatic activity and big size. Therefore, the special features of these ARV structures, such as their size, organization and the possibility to target them with foreign antigens, made them a potential candidate for a vaccine platform.

## 3. Interaction of Nano- or Microcarriers with the Immune System

Depending on their route of administration, nano- and microparticles first encounter innate immune cells such as macrophages or dendritic cells (DCs) (Figure 2a). Antigen-presenting cells (APCs), such as macrophages, B-cells or DCs—the most potent APCs, assist in the progression from innate to adaptive immune responses. Uptake of multimeric particles by APCs produces their activation via pattern recognition receptors (PRRs)—such as Toll-like receptors (TLRs) or C-type lectin receptors (CLRs)—which leads to a pro-inflammatory response and subsequent particle internalization, antigen processing and presentation [102,103]. Therefore, microparticles or nanoparticles such as VLPs can promote the activation of DCs and the presentation of antigens by MHC class I or II molecules, eventually leading to priming of CD8+ and CD4+ responses. The plasticity of nano- and microdelivery platforms allows the passive targeting of APCs for optimizing internalization efficiency and, eventually, maximizing the elicited immune response. According to the particle rigidity, shape, surface charge and size, differential biodistribution and particle uptake have been observed [104,105]. Immunogenicity is highly impacted by the draining properties of the antigenic particle [106]. Nanoparticles of less than 200 nm in diameter distribute more rapidly to lymph nodes whereas nanoparticles of more than 500 nm and microparticles are taken up at the site of injection by tissue-resident DCs and macrophages, and actively transported to the lymphatic tissues. Thereby, through the interstitial fluid and the collecting lymphoid vessels, the smaller particles (<50 nm) can reach lymphoid organs (such as the spleen or lymph nodes) more efficiently and interact with follicular DCs and resident T and B cells [105,107]. Moreover, DCs preferentially internalize small size (virus-size) particles [108,109,110,111]. Similarly, macrophage uptake seems to be also constrained by particle size and surface charge [104,112,113]. In any case, active targeting can significantly optimize DCs and macrophage internalization efficiencies [111,114].

Induction of both arms of the adaptive immune system, cell-mediated and antibody-mediated responses, is key for the clearance of viral pathogens and has been attained by using nano- or microcarrier systems (Figure 2b). After particle uptake, antigenic proteins delivered as VLPs or by any other carrier system are processed and antigenic peptides are presented with similar efficiency on MHC-I or MHC-II molecules by APCs to trigger the activation of CD8+ T cells or Th CD4+ cells, respectively [115,116]. These technologies induce a potent stimulation of both Th1 and Th2 responses [117]. In some cases, microparticles and nanoparticles with sizes >500 nm show a tendency to induce Th2 responses whereas particles of <100 nm seem to induce Th1-biased responses [111]. Exogenous antigens, such as those present in VLP or other particulate systems, are usually presented by MHC-II molecules but can bind to MHC-I molecules too [118]. Cross-presentation—or cross-priming—which is the presentation of antigenic peptides from exogenous proteins on MHC class I molecules by APCs, is an essential step for the initiation of CD8+ T-cell responses [119]. Cytotoxic T-lymphocyte (CTL) responses are of vital importance for the clearance of viral pathogens, such as BTV, AHSV and EHDV [120]. Expansion of CD8+ T cells has been observed in BTV- and AHSV-infected and vaccinated hosts [121,122]. Indeed, infected animals can mount CD8+ T-cell-mediated protective responses against homologous and heterologous non-related serotypes of BTV and AHSV—with highly conserved proteins VP7, NS1 and NS2 as potential targets—in the absence of neutralizing antibodies (Nabs) [123,124,125,126,127,128,129,130,131]. In this regard, DC cross-priming of cytotoxic T cells plays a critical role after VLP vaccination [132]. Compared to soluble antigens, antigen delivery by nano- or microparticles promotes endosomal scape through a variety of mechanisms (reviewed in [133]) after uptake of these structures by professional DCs, thus enhancing the access of exogenous antigen to the classic MHC class I pathway and the subsequent T-cell cross-priming and activation. However, particle size influences the cross-presentation pathway for a given antigen. In this sense, MHC class I-peptide complexes formation yields could differ between the ‘phagosome-to-cytosol’ and the direct endosomal loading pathways, which could explain the higher efficacy of nanoparticles to induce MHC class I cross-presentation [134,135,136].

Although orbiviruses can transiently impair B-cell responses [137], humoral immunity is also triggered after infection with BTV, AHSV or EHDV, with the most exposed protein of the virion and major determinant of virus serotype, VP2, as the main target of Nabs [138,139,140,141]. Th2 cell development subsequently leads to clonal expansion of B cells and affinity maturation of produced antibodies [142,143]. The immunoglobulin class switch toward IgG is promoted by the interaction of Th2 cells with activated B cells. Particulate systems, like VLPs, are strong stimulators of B-cell-mediated immune responses. These highly organized and dense antigenic structures prompt direct cross-linking with the BCR and activation of B cells, which leads to particle internalization, B-cell proliferation and an increased expression of MHC-II and co-stimulatory molecules [103]. Moreover, due to the structural repetitiveness of these nano- and microcarrier systems, stimulation of T-cell-independent humoral immune responses based on IgM and IgG has been observed [144]. Not just that, VLPs and other micro- or nanocarriers also have the potential to induce CTL responses independently of CD4+ Th cells and are able to induce long-lived, antibody-secreting plasma cells and memory B and T cells [144,145,146,147,148,149,150,151,152].

## 4. Bluetongue Virus (BTV)

Bluetongue virus (BTV) is the type species of the genus *Orbivirus* and causative agent of bluetongue (BT), a hemorrhagic ruminant disease with a broad host range, although mortality and morbidity varies among species [153,154]. This disease is mainly appreciated in sheep and some wild ruminants such as white-tailed deer (*Odocoileus virginianus*), with significant mortality rates. On the contrary, goat, cattle and most wild ruminants usually display either subclinical or asymptomatic BT signs [155,156,157]. Traditionally, BTV global distribution is enclosed to regions between approximately 50° N and 35° S. Nonetheless, northwards expansion has occurred during recent years, with it being identified in all continents except Antarctica [5]. The worldwide economic impact of BT is estimated to be 3 billion US dollars per year [158]. Such expenses are the result of productivity losses, animal death, animal trade restrictions and the implementation of surveillance measures [158,159]. Thus, it is included in the notifiable disease list of the World Organization for Animal Health (WOAH, former OIE) because of its detrimental consequences in the livestock industry. More than 29 serotypes have been identified so far [17], based on extensive phylogenetic studies, sequencing data and cross-neutralization assays [18,160]. The global distribution of individual BTV serotypes is highly heterogenous [161]. Conventional approaches based on inactivated or live attenuated vaccines have been permitted to contain BTV. However, they present several pitfalls, highlighting their serotype specificity and lack of DIVA properties. A huge effort has been made to develop next-generation vaccines that could solve drawbacks of these marketed classical vaccines, including subunit, viral vector, DISC (disabled infectious single cycle) and DISA (disabled infectious single animal) vaccines [17,18].

### 4.1. Virus-Like Particles (VLPs)

VLPs and CLPs (core-like particles) have been extensively evaluated as potential nanoparticle-based vaccines against BTV (Table 1). Homogenous BTV CLPs and VLPs were first assembled in baculovirus–insect cell expression systems. CLPs consist on the simultaneous expression of VP3 and VP7 in absence of the polymerase complex (VP1, VP4 and VP6) and any non-structural protein, being identical in size and shape to authentic BTV cores [162]. VLPs are composed of the core particle (VP3 and VP7) and the outer capsid layer (VP2 and VP5), and resembled BTV in size, appearance and biochemical constitution, lacking the double-stranded RNA and the RNA polymerase complex (Figure 3) [138]. Importantly, VP3 and VP7 are relatively well conserved between serotypes so that outer capsid proteins from different serotypes can be coated onto heterologous inner core proteins, which implies an advantage of this VLP system to generate vaccines to new serotypes by replacing VP2, the most variable BTV protein [163]. Apart from a baculovirus expression system, a plant-based high-level expression system was also implemented to produce assembled subcore-, core- and virus-like particles of BTV [164,165].

Regarding their evaluation as vaccine candidates, CLPs showed immunogenicity in vivo but were incompletely protective against BTV [138,163,164,166]. The major outer core protein and cross-reactive antigen VP7 has been shown to be protective against BTV [167,168]. So, despite CLPs lack of major immunogenicity determinant VP2, the main inductor of neutralizing antibodies (NAbs), the induction of a cell-mediated immune response specific of VP7 could explain the mild protective capacity observed [166]. Conversely, VLPs were able to confer serotype-specific protection from clinical disease and viral replication in sheep, one of the natural BTV hosts, even in the long-term [72,166,169,170,171,172]. Nonetheless, complete protection against BTV was dependent on the utilization of adjuvants, as adjuvant-free VLPs were demonstrated to be only partially protective in sheep, inducing low levels of NAbs [172]. As stated above, one of the major drawbacks of BTV vaccination is the serotype-specificity. Cross-protection was only partially achieved against phylogenetically related serotypes after immunization with VLPs [169]. In consequence, the rationale would be the design of a cocktail of VLPs of different serotypes that could work as a multivalent vaccine. This strategy has been successfully explored as multivalent vaccine candidates protected against homologous serotypes [72,169,171]. Considering that development of multivalent vaccines could induce immune interferences between different antigens present within the same vaccine formulation [173,174,175], it is important to note that the combination of VLPs containing VP2 from different serotypes did not lead to interferences of any kind, which places cocktails of VLPs as potential DIVA, multiserotype and inherently safe vaccines against BTV. Nevertheless, these vaccine candidates have not been marketed yet, despite being proven to be effective, safe and DIVA candidates, something that could be explained by a low affordability. In a similar way, plant-produced VLPs offered protective immunity against homologous serotypes [164,165] so that VLP production in plants may work as a more cost-effective alternative with the advantage of a feasible scalability. Leastways, a plausible and more likely affordable alternative to BTV VLPs is the 60-meric self-assembled enzymatic complexes made of lumazine synthase from the hyperthermophile *Aquifex aeolicus* carrying the immunodominant VP2 [176]. These virus-like nanoparticles have been applied for host receptor identification during virus entry, but their potential as vaccines candidates against BTV has not been characterized so far.

One of the exploitations of VLPs is their use as carriers for foreign epitopes [177,178,179,180,181,182]. The rabbit hemorrhagic disease virus VP60 VLPs carrying a six-residue epitope from BTV virus capsid protein VP7 is an example. This vaccine candidate was not further evaluated as a vaccine candidate [183]. BTV CLPs have been also explored as a delivery platform. These were loaded on their surface with heterologous antigens such as preS2 of the hepatitis B virus [184], an immunogenic epitope of the rabies virus glycoprotein [185], a T-cell epitope of the influenza A virus matrix (M1) protein [186] or a sequence of the bovine leukemia virus (BLV) gp51 protein [187]. The EIII ectodomain of dengue virus serotypes 1 (DENV1) and 4 (DENV4), and Zika virus, have been also displayed on the inner surface of BTV CLPs fused to the amino terminal end of VP3 [188]. Although these strategies showed some mild immunogenicity to foreign antigens, the length of the aminoacidic inserted sequence dramatically conditioned the assembly and stability of chimeric CLPs in some cases. The interior of empty plant-based BTV CLPs has also been labeled with GFP molecules [189].

### 4.2. Avian Reovirus µNS- Microspheres Carrying BTV Antigens

Recently, different studies have noted the capability of the conserved non-structural protein NS1 of BTV to promote a specific CD8+ T-cell response able to protect immunized animals against several non-related serotypes of BTV [124,127]. The combination of both virus-neutralizing antibodies and cytotoxic T lymphocytes (CTLs) is crucial for the development of a long-lasting immunity in animals infected with BTV [120,190]. The immune response induced by BTV VLPs is eminently based on the induction of NAbs, although the presence of VP7 may also stimulate a cross-reactive cell-mediated response. Immunization with NS1, VP2 and VP7 loaded on avian reovirus muNS-microspheres fostered both arms of the adaptive immune response, providing sterile immunity against homologous BTV serotype in IFNAR(-/-) mice [191] (Table 1), mainly mediated by triggering neutralizing antibodies and CD4+ T-cell activation. Heterologous combination with a modified vaccina Ankara (MVA) viral vector expressing these BTV antigens led to complete heterologous protection against BTV, which could rely on a more potent BTV-specific cell response influenced by the viral vector used [192]. In addition, BTV-muNS-MS presented an intrinsic adjuvant capacity, which differs from VLPs adjuvant-dependent response. Nonetheless, results on the efficacy of BTV-muNS-MS in the mouse model must be assayed in a BTV natural host. In any case, due to the plasticity and easy method of production implied in this technology, additional BTV antigens could be included in this subunit vaccine formulation, as could be the case of NS2 or its amino-terminal end (NS2-Nt), where combination with NS1 optimizes the multiserotype protection elicited by NS1 alone, avoiding the progression of clinical disease in sheep [125].

**Table 1 vaccines-10-01124-t001:** Nanoparticle- and microparticle-based vaccine candidates generated against BTV.

Vaccine Type	Antigen Included	Dose	Animal Model	Challenge	Adjuvant	Immunogenicity and/or Protection	Ref.
CLP ^a^	VP7, VP3	Not evaluated as vaccine candidate	-	-	-	-	[162]
CLP ^a^	VP7, VP3	Four doses (75 µg per dose)	Guinea pig	Not challenged	Adjuvant-free	Absence of Nabs. Protection not Assessed	[138]
VLP ^a^	VP2 (BTV-10), VP5, VP7, VP3					Neutralization against BTV-10. Protection not Assessed	
VLP ^a^	VP2 (BTV-1 or BTV-17), VP5, VP7, VP3	Four doses (100 µg per dose)	Guinea pig	Not challenged	Incomplete Freunds’ adjuvant	Neutralization against BTV-1 or BTV-17. Protection not Assessed	[163]
VLP ^a^	VP2 (BTV-10), VP5, VP7, VP3	Two doses (10, 50, 100 or 200 µg per dose)	1-year-old Merino sheep	BTV-10	Adjuvant-free	Neutralization against BTV-10. Partial Homologous protection ^c,d^.	[172]
					IFA		
					AI(OH)3		
					ISA-50	Neutralization against BTV-10. Homologous protection ^c,d^.	
VLP ^a^	VP2 (BTV-10 or BTV-17), VP5, VP7, VP3	Two doses (10 or 50 µg per dose)	1-year-old Merino sheep	Homologous (BTV-10 or BTV-17)	ISA-50	Neutralization against BTV-10 and BTV-17. Partial cross-neutralization against BTV-4. Homologous protection ^c,d^.	[169]
	VP2 (BTV-10 and BTV-17), VP5, VP7, VP3			Heterologous (BTV-4 or BTV-11)		Neutralization against BTV-10 and BTV-17. Partial cross-neutralization against BTV-4. Partial heterologous protection ^c,d^.	
	VP2 (BTV-1, BTV-2, BTV-10, BTV-13 and BTV-17), VP5, VP7, VP3	Two doses (50 µg per dose)		Homologous (BTV-13) and Heterologous (BTV-16)		Neutralization against BTV-10, BTV-13 and BTV-17. Homologous protection ^c,d^. Mild heterologous protection ^c,d^.	
VLP ^a^	VP2 (BTV-2), VP5, VP7, VP3	Two doses (10 or 20 µg per dose)	10-month-old cross-bred Pre-Alpes sheep	Homologous (BTV-2)	SEPPIC	Neutralization against BTV-2. Homologous protection ^c,e,f^.	[170]
VLP ^a^	VP2 (BTV-1), VP5, VP7, VP3	Two doses (unspecified dose)	7–8-month-old male Merino sheep	Homologous (BTV-1)	SEPPIC	Neutralization against BTV-1. Homologous protection ^c,e,f^.	[171]
	VP2 (BTV-1 and BTV-4), VP5, VP7, VP3			Homologous (BTV-1 and BTV-4)		Neutralization against BTV-1 and BTV-4. Protection against BTV-1 ^c,e,f^. Partial Protection against BTV-4 ^c,e,f^.	
CLP ^a^	VP7, VP3	Two doses (~50–100 µg per dose)	18-month-old female Karagouniko cross-bred sheep	Homologous (BTV-1)	SEPPIC	Low protection ^c,d,e,f^.	[166]
VLP ^a^	VP2 (BTV-1), VP5, VP7, VP3					Neutralization against BTV-1. Homologous protection ^c,d,e,f^.	
VLP ^a^	VP2 (BTV-8), VP5, VP7, VP3	Two doses (20 µg per dose)	10-month-old cross-bred Pre-Alpes sheep	Homologous (BTV-8)	SEPPIC	Neutralization against BTV-8. Homologous protection ^c,e,f^.	[72]
	VP2 (BTV-8, BTV-1 and BTV-2), VP5, VP7, VP3	Two doses (60 µg per dose)				Neutralization against BTV-1, BTV-2 and BTV-8. Homologous protection ^c,e,f^.	
CLP ^b^	VP7, VP3	Two doses (200 µg per dose)	1-year-old Merino sheep	Homologous (BTV-8)	SEPPIC	No protection ^c,e^.	[164]
VLP ^b^	VP2 (BTV-8), VP5, VP7, VP3	Two doses (50 µg per dose)				Neutralization against BTV-8. Homologous protection ^c,e^	
VLP ^b^	VP2 (BTV-2 or BTV-4), VP5, VP7, VP3	Two doses (~15–30 µg per dose	6–12-month-old Merino sheep	Not challenged	SEPPIC/Alhydrogel	Neutralization against BTV-3 and BTV-4. Protection not assessed	[165]
Lumazine synthase complexes ^a^	VP2	Not evaluated as vaccine candidate	-	-	-	-	[176]
Carrier RHDV VLP ^a^	Six residue epitope of VP7 (BTag)	Not evaluated as vaccine candidate	-	-	-	-	[183]
muNS-Mi ^a^	NS1, VP2 (BTV-4), VP7	Two doses (150 µg per dose)	Adult IFNAR(-/-) mice	Homologous (BTV-4) and heterologous (BTV-1)	Self-Adjuvant	Neutralization against BTV-4. Cell immune responses specific of VP7 or NS1. Homologous protection ^g,c,d^. Partial heterologous protection ^g,c,d^.	[191]
muNS-Mi/MVA ^a^	NS1, VP2 (BTV-4), VP7	Two doses (prime: 150 µg; boost: 10^7^ PFU)	Adult IFNAR(-/-) mice	Homologous (BTV-4) and heterologous (BTV-1)	Self-Adjuvant	Neutralization against BTV-4. NS1-spcecific CD8+ T-cell response. Homologous and heterologous protection ^g,c,d^.	[192]

^a^ Recombinant baculovirus expression system; ^b^ plant-based expression system; ^c^ lessened clinical signs; ^d^ reduction or absence of viraemia; ^e^ steady rectal temperatures; ^f^ reduction or absence of RNAmia; ^g^ increased survival rate.

### 4.3. A Self-Assembled Nanoparticle Vaccine Platform Based on BTV-NS1 Tubules

A feature of BTV infection is the generation of tubular structures of ~52.3 nm in diameter and ~100 nm in length within the cytoplasm of infected cells [193], originated from the non-structural protein 1 (NS1) multimerization. The 64 kDa NS1 protein, which acts as a positive regulator of viral protein synthesis [194], is the most abundant viral protein expressed in the cytoplasm of BTV-infected cells and is the only one responsible for the formation of these characteristic dynamic helical tubules [193,195,196,197]. Akin to the coat of a filamentous phage or tobacco mosaic virus displaying foreign antigens on their surface [198,199,200,201,202,203], these NS1 multimeric structures have been proposed as an efficient particulate delivery system as long as they are easily purified, highly stable at high temperatures (between 30 and 45 °C) and at a pH of 7 to 8.5, can accommodate bigger polypeptides than VLPs (up to 552 amino acids [204]) and are composed of thousands of NS1 monomers, which would ensure a high exposure of the foreign antigen [193,204,205,206]. The C-terminus of the NS1 protein is regularly disposed on the surface of the tubules [197,207,208]. The addition of foreign epitope sequences (as fusion proteins or by cross-linking [209]) in the C-terminal end of NS1 did not interfere with tubule formation and allowed the exposure of antigenic sequences in the surface of NS1 tubules [207,208]. Interestingly, these chimeric structures showed immunogenicity against inserted antigenic sequences, successfully stimulating both cellular and humoral arms of the immune system in vivo [204,209,210,211,212] and showing a significant degree of protection against the pathogen of interest in some cases [210,211].

Considering recent studies that corroborate the capacity of NS1 to efficiently protect against BTV [124,125,127], it must be noted that the NS1-based VLP-like carrier has not been evaluated yet as a vaccine candidate against BTV. In view of the advantages of this technology, it is a question that needs to be addressed. If so, taking into consideration the proficiency of these NS1 tubules to simultaneously carry a mix of different foreign antigenic sequences [208], it would be feasible to delineate multivalent vaccines against different ruminant viral diseases. This is an important issue when considering a reduction in the time and costs of production, and faster and more effective vaccination campaigns. Alike to some evaluated multivalent viral vector vaccines [213,214,215,216,217], delivery of antigens from the Rift Valley fever virus (RVFV), lumpy skin disease virus (LSDV), peste des petits ruminants virus (PPRV) or rinderpest virus could lead to multivalent vaccine candidates against some of these ruminant pathogens including BTV.

## 5. African Horse Sickness Virus (AHSV)

African horse sickness is on the WOAH list of notifiable viral diseases and is one of the most lethal viral diseases that affect equids [4]. Mortality rates can reach up to 90%, which depends on the form of the disease and the presence of naturally resistant and/or previously infected horses [218]. Asiatic and European donkeys also suffer severe forms of AHS [218]. Other equids species, such as some zebra species and the African donkey, suffer the less severe form and act as primary maintenance reservoirs [219,220]. AHSV is endemic in regions of Africa, and causes epidemics in Europe and Asia [218]. Without forgetting working equids in low-income countries, AHS is a major economic threat to the high-performance horse production industry [4]. AHSV outbreaks are controlled by the quarantine of animals coming from endemic areas, stabling and vaccination [4]. Vaccination with live attenuated strains of AHSV is the unique available prophylactic measure to control this viral disease. The currently used LAV contains serotypes 1, 2, 3, 4, 6, 7 and 8. However, drawbacks shared with BTV LAVs, such as documented gene segment re-assortment between outbreak and vaccine strains and reversion to virulence of attenuated vaccine strains [221], plus the need of multiple vaccination doses over the years [222], urge the evaluation of alternatives. In this sense, inactivated, DNA, subunit, viral-vectored and reverse genetics-based vaccines, have been explored [19,223].

Analogous to BTV, co-expression of major core structural proteins VP3 and VP7 by recombinant baculovirus expression systems led to the generation of AHSV CLPs that resembled in size and appearance genuine AHSV cores, as demonstrated by electron microscopy [224]. Later, Maree et al. [225] reported the production of AHSV-9 VLPs by baculovirus-mediated co-expression of the four major structural proteins (VP2, VP5, VP7 and VP3) in insect cells (Table 2). Importantly, the authors also showed the molecular interaction of VP2 or VP5 with CLPs, demonstrating that the simultaneous expression of both proteins is not imperative for individual attachment of these proteins to CLPs, resulting in what could be called as partial VLPs, composed of VP2 or VP5 along with VP7 and VP3. Nonetheless, partial VLPs, carrying either VP2 or VP5, have not been evaluated yet for their possible immunogenic potential. The authors also reported the formation of crystalline structures made of VP7, a phenomenon also observed during AHSV infection [226], and hypothesized them as the cause of a low VLP formation yield. To address this problem, rational design of a mutant of the highly hydrophobic VP7 resulted in an increased solubility and improved efficiency of CLPs (and VLPs as a consequence) production. Similarly, a mutated version of VP7 (seven amino acid substitutions in the N-terminal region) was designed to increase VLP concentration in *Nicotiana benthamiana* plant expression systems, in which fully assembled AHSV-5 VLPs were successfully expressed. Density gradient-purified VLPs showed immunogenicity in guinea pigs–VP2, VP5 and VP7 were recognized by sera from immunized-with NAbs titers ranging from 1:640 to 1:5120 against the homologous AHSV-5 and, importantly, with titers between 1:56 and 1:160 against the heterologous AHSV-8 [227]. Surprisingly, VP7-mutant VLPs were excluded for immunization of horses as authors exposed that wild-type-VP7 VLPs were purified in a similar yield to that of the mutated protein. In any case, AHSV-5 VLPs did induce high titers of NAbs in the high (200 µg; 1:320) and low (100 µg; 1:160) dose immunization groups against the homologous AHSV-5. Like guinea pigs, immunized horses also developed a strong heterologous neutralizing response against AHSV-8 (between 1:160 and 1:320). Two horses also reached a AHSV-4 neutralizing antibody titer of 1:112 after the booster dose although it might be a product of an anamnestic response due to previous exposure to the LAV [228]. Agroinfiltration of *Nicotiana benthamiana* dXT/FT plants was also exploited for production of immunogenic chimeric AHSV VLPs by exchanging VP2 proteins of different serotypes [70].

Presumably, the former reviewed strategies are serotype-specific and would require the combination of a cocktail of VLPs for design of multivalent vaccines as for BTV. One of the virus-encoded nonstructural proteins, NS1, is highly conserved between the different serotypes. In addition, this protein, which forms tubular structures in the cytoplasm of infected cells similar to BTV NS1 [229], contains T CD8+ epitopes within its aminoacidic sequence [230]. For those reasons, ARV muNS-MS were loaded with AHSV-4 NS1 (Table 2). Immunization of IFNAR(-/-) mice with two doses of muNS-Mi-NS1 elicited a T-cell-mediated immune response that partially protected against the lethal challenge with the homologous AHSV-4. Much better, heterologous combination with MVA expressing NS1 conferred sterile protection in IFNAR(-/-) mice. Despite AHSV-9 not being lethal in IFNAR(-/-) mice, the heterologous strategy also prevented mice from developing viraemia and weight loss, indicating the multiserotype protective potential of NS1 [128]. Certainly, further research should be carried out on AHSV natural hosts to confirm the results observed in the mouse model.

## 6. Epizootic Hemorrhagic Disease Virus (EHDV)

EHD, caused by EHDV, is a non-contagious hemorrhagic disease included in the WOAH list of multispecies/transboundary diseases. EHD is historically associated with wild and domestic ungulates such as white-tailed deer and cattle, and, especially with white-tail deer, showing important mortality rates [20]. Notably, EHDV infection in cattle has led to clinical cases, causing morbidity rates varying from 1% to 18% whereas sheep and goats suffer a subclinical infection and may play a role in EHDV epidemiology [20,231,232]. This virus has been detected in regions of North, Central and South America, Australia, Asia and Africa [231,232,233,234,235,236,237,238,239,240]. To date, there is no evidence of the presence of EHDV in Europe. However, its presence in this continent is feasible as EHDV has been detected in countries of the Mediterranean Basin such as Morocco, Algeria, Tunisia, Israel, Jordan and Turkey and this virus shares epidemiologic similarities with BTV [20]. The economic losses derived from EHDV infection of dairy cattle has been estimated in 2.5 million US dollars due to reduced milk and meat production during an outbreak in 2006 in Israel [241]. Thus, the presence of susceptible and naïve cattle in Europe makes an outbreak of EHDV a likely high-risk upcoming event.

The effort to develop next-generation vaccines against EHDV is not comparable to those made for the related orbiviruses BTV and AHSV. Up to seven distinct serotypes of EHDV have been identified so far, and, with the exception of autogenous vaccines used in the US, just a monovalent live attenuated vaccine and a bivalent (Ibaraki virus (EHDV-2)/bovine ephemeral fever virus) inactivated vaccine are commercialized and used in Japan [242]. Together with them, a recombinant subunit vaccine is the unique experimental candidate that has demonstrated efficacy in the natural host. Baculovirus-expressed VP2 induced a NAbs response in cattle and white-tail deer, and conferred sterile serotype-specific protection against EHDV challenge [243]. Recently, a reverse genetic system has been developed for EHDV, which could serve for novel EHDV vaccine generation [244,245]. In parallel, the knowledge gathered for BTV VLPs design and production was applied for the generation of EHDV CLPs and VLPs (Table 2). As previously mentioned for BTV and AHSV, CLPs of EHDV are composed of the intermediate VP7 and the innermost VP3 proteins that act as scaffolds for the assembly of VP2 and VP5, the most variable proteins of EHDV, to generate VLPs. Alshaikhahmed and Roy [246] described the generation of CLPs and VLPs of serotype 1 of EHDV by using the baculovirus expression system. Rabbit immunization with these VLPs demonstrated their capacity to promote the induction of strong titers of homologous NAbs in presence of adjuvants. Cell-mediated immune responses were not evaluated. Low levels of NAbs against EHDV-1 related heterologous serotypes (EHDV-2 and EHDV-6) were also detected. They also showed that EHDV-1 CLPs can serve as scaffolds for the generation of VLPs of other EHDV serotypes, which relies on the high conservation degree of VP3 and VP7 [233,247]. Similarly, Forzan et al. [248] successfully explored the usage of the baculovirus expression system for the generation of field isolate EHDV-6 VLPs, although there is a lack of data on related immunogenicity and protection. Importantly, recombinant baculovirus EHDV VLPs production may be restrained by the same conditionings previously discussed for BTV VLPs. Therefore, plant-based production would be highly desirable.

**Table 2 vaccines-10-01124-t002:** Nanoparticle- and microparticle-based vaccine candidates designed against AHSV and EHDV.

Species	Vaccine Type	Antigen Included	Dose	Animal Model	Challenge	Adjuvant	Immunogenicity and/or Protection	Ref.
African horse sickness virus (AHSV)	CLP ^a^	VP7,VP3	Not evaluated in animal model	-	-	-	-	[224]
	CLP ^a^	VP7,VP3	Not evaluated in animal model	-	-	-	-	[225]
	Partial VLP ^a^	VP2 (AHSV-9),VP7,VP3						
		VP5,VP7,VP3						
	VLP ^a^	VP2 (AHSV-9),VP5,VP7,VP3						
	VLP ^b^	VP2 (AHSV-5),VP5,VP7,VP3	Two doses (Prime: 16.5 µg; Boost: 50 µg)	Guinea pig	Not challenged	5% Montanide PET Gel A adjuvant	Neutralization against AHSV-5. Partial cross-neutralization against AHSV-8.	[227]
			Two doses (100 or 200 µg per dose)	Horse	Not challenged		Neutralization against AHSV-5. Cross-neutralization against AHSV-8.	[228]
	VLP ^b^	VP2 (AHSV-6), VP5,VP7,VP3	Two doses (~200 µg per dose)	6–12 months foals	Not challenged	5% Montanide PET Gel A adjuvant	Neutralization against AHSV-6.	[70]
	muNS-Mi ^a^	NS1 (AHSV-4)	Two doses (50 µg per dose)	Adult IFNAR(-/-) mice	Homologous (AHSV-4)	Self-adjuvant	NS1-specific CD8+ T-cell response. Partial homologous protection ^c,d,e^.	[128]
	muNS-Mi/MVA ^a^		Two doses (prime: 50 µg; boost: 10^7^ PFU)		Homologous (AHSV-4) and heterologous (AHSV-9)		NS1-specific CD8+ T-cell response. Sterile homologous and heterologous protection ^c,d,e^.	
Epizootic hemorrhagic disease virus (EHDV)	CLP ^a^	VP7, VP3	Two doses (prime: 500 µg; boost: 250 µg)	Rabbit	Not challenged	Incomplete Fruend’s adjuvant	Induction of VP3- and VP7-specific antibodies.	[246]
	VLP ^a^	VP2 (EHDV-1),VP5,VP7,VP3	Two doses (prime: 500 µg; boost: 250 µg)	Rabbit	Not challenged	Incomplete Fruend’s adjuvant	Neutralization against EHDV-1. Mild neutralization against EHDV-2 and EHDV-6.	
	VLP ^a^	VP2 (EHDV-6),VP5,VP7,VP3	Not evaluated in animal model	-	-	-	-	[248]

^a^ Recombinant baculovirus expression system; ^b^ plant-based expression system; ^c^ increased survival rate; ^d^ reduction or absence of viremia; ^e^ reduced body weight loss.

New EHDV vaccine design strategies must be explored in the future. Aside from VP2, the study of alternative EHDV antigens should be carried out for the development of multiserotype protective responses, similar to BTV or AHSV [125,127,128,230,249]. Some technologies depicted here are suitable for that purpose. Moreover, validation of a laboratory animal model for EHDV infection would immensely ease the screening of new EHDV vaccination alternatives. IFNAR(-/-) mice, widely used for BTV and AHSV vaccine evaluation [250,251,252], are susceptible to EHDV, showing a dose-dependent lethality [253], and might facilitate the study of EHDV pathogenesis and the development of new effective vaccines for EHDV.

## 7. Conclusions

Antigen delivery by nano- or microparticles is one of the most explored approaches for vaccine development against infectious diseases. Although BTV and AHSV have been controlled through vaccination with LAVs and inactivated vaccines specific of virus serotype, they still continue to be livestock-threatening diseases. Moreover, other orbiviruses, as in the case of EHDV, reunite important considerations as threating pathogens for the livestock industry and wildlife in non-endemic areas. Here, we have reviewed a wide range of nanoparticle- and microparticle-based vaccines that can offer a solution against these three orbiviruses. Although nano- or microparticle vaccines against orbiviruses can be improved, they offer efficacy, safety, affordability and DIVA capacity, the main vaccine standards.

## Figures and Tables

**Figure 1 vaccines-10-01124-f001:**
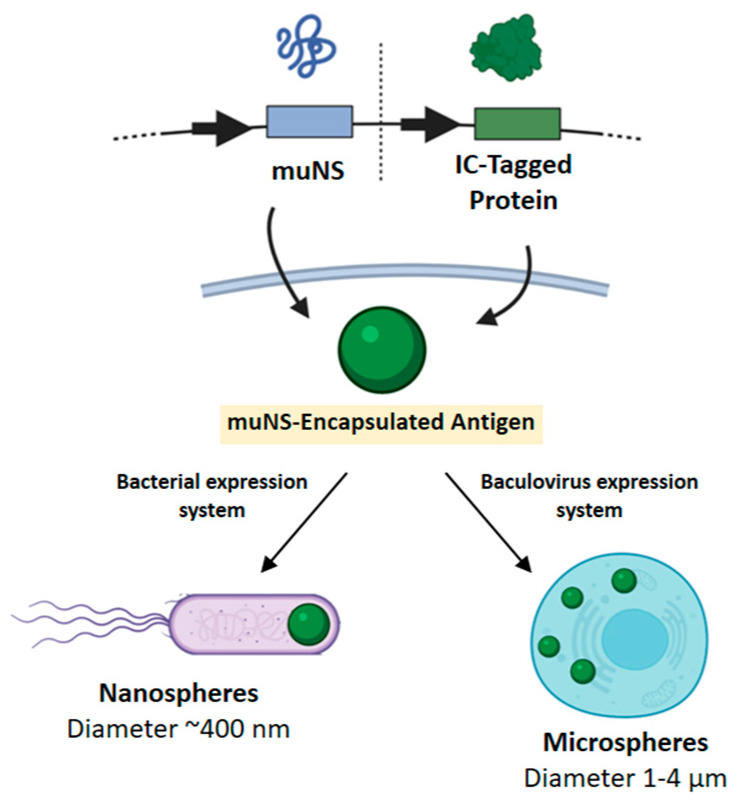
IC-Tagging methodology allows for incorporation of a given antigen into muNS particles. Bacterial expression leads to self-adjuvant epitope-loaded nanospheres of ~400 nm whereas baculovirus-based expression produces microspheres with diameters between 1-4 μm.

**Figure 2 vaccines-10-01124-f002:**
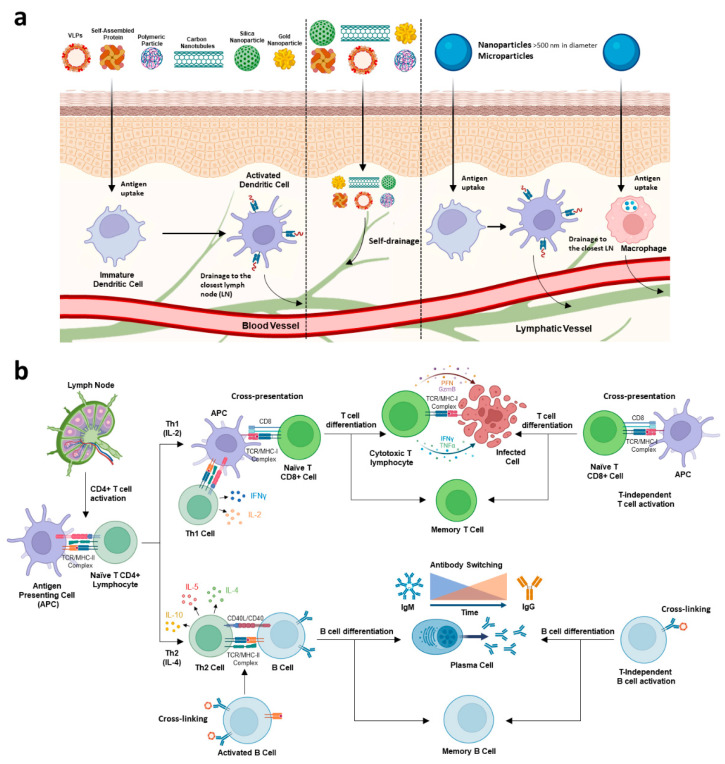
Nano- and microparticle delivery systems induce both humoral and cellular immune responses. (**a**) Particle uptake by antigen-presenting cells (APCs) is a crucial step for induction of potent immune responses. Depending on a variety of factors, nanoparticles can be internalized by APCs or directly reach lymphatic tissue through lymphatic vessels. Microparticles are phagocytosed by APCs. (**b**) After antigen presentation by APCs, CD4+ T cells differentiate into Th1, promoting the generation of cytotoxic and memory T cells by cross-presentation, or Th2, enhancing the differentiation of B cells into plasmatic or memory B cells. Nano- and microparticles can also promote T- and B-cell activation independent from Th cells.

**Figure 3 vaccines-10-01124-f003:**
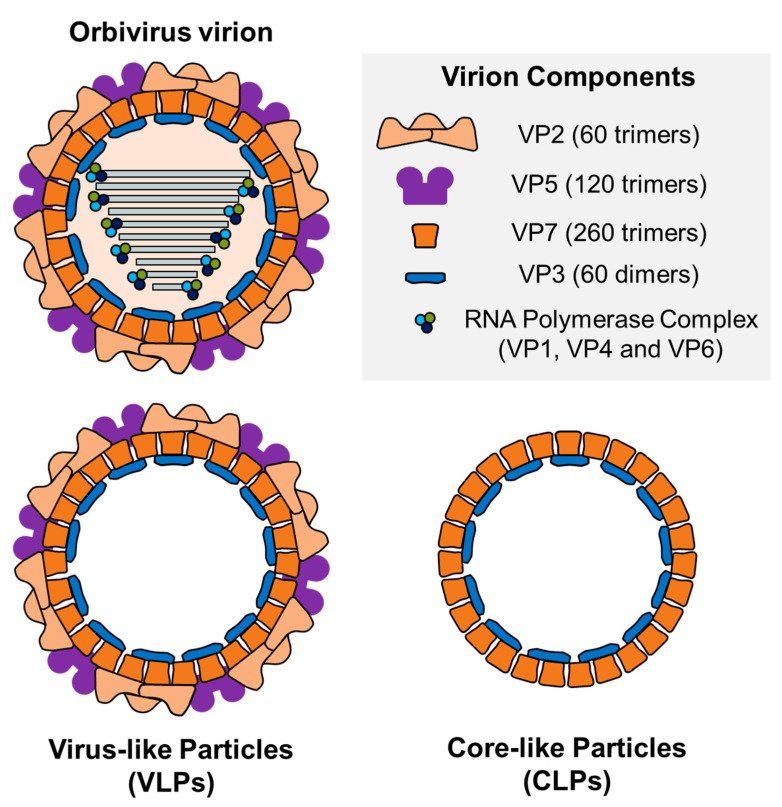
Diagrammatic representation of a section of an orbivirus virion, a virus-like particle (VLP) and a core-like particle (CLP). 60 trimers of VP2 and 120 trimers of VP5 constitute the outer capsid of the virion. The core is composed of the intermediate protein layer, comprised of 260 trimers of VP7, and the inner capsid formed by 60 dimers of VP3. Inside the core, VP1, VP4 and VP6 constitute the RNA polymerase complex. VLPs lack genetic material as well as the RNA polymerase complex components. CLPs are protein assemblies just formed by VP7 and VP3.

## Data Availability

Not applicable.

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
