# Peer review of "Nanoparticle- and Microparticle-Based Vaccines against Orbiviruses of Veterinary Importance"

_vaccines, 2022, doi:10.3390/vaccines10071124_

Round 1

Reviewer 1 Report

The review article focuses on the current nanoparticle and microparticle-based vaccine approaches to prevent infection with clinically relevant orbiviruses including bluetongue virus (BTV), African horse sickness virus (AHSV), and epizootic hemorrhagic disease virus (EHDV). The authors go into great detail about the types of nano and microparticle vaccines and how an immune response is generated. Then, the review looks specific at the vaccines for BTV, AHSV, and EHDV summarizing each section in a table. Some of the approaches include VLP and CLP-like technologies utilized for protection and conferred immunity from these viruses and other serotypes. The review addresses novel approaches to vaccination such as assessing the efficacy of a BTV-NS1 based presentation of antigens through use of novel NS1-multimeric-microtubules that are characteristic of infection of some of these orbiviruses. Overall, the review is comprehensive and provides insight into potential vaccine designs that should be tested but text clarification and additional figures are needed to support the manuscript.

Major Comments:

1.       Figure 1 does not support the content of the review. It would be better suited to present the organization of the genera and families as a phylogenetic tree with the viruses and serotypes that are discussed included.

2.       An additional figure with an illustration of a cross-section of the orbivirus virion would provide a better reference for the readers for the proteins that are expressed to produce the VLPs and CLPs.

3.      Much of the second half of section 2, describing the immune response to nano- and microparticles, is primarily describing a generic immune response. The authors should summarize the more common concepts and include information on the protective immune response required for orbivirus infections.

Minor Comments:

1.       The in-depth description of nanoparticle and microparticle vaccines in the introduction (lines 68-155) should be reworked into a new section with subsections for each vaccine approach. This will enhance the flow and readability of the review.

2.       In line 211, it is unclear why the authors described immunoglobulin class switching (or isotype switching) as “so-called antibody switching”. Please adjust.

3.       Please include the endemic regions of BTV.

4.       For Table 1 and 2, please include how many doses of the vaccine were administered and elaborate on how protection was assessed, e.g., survival, viral burden, etc. If space is limiting, the expression system can be denoted as a symbol on the vaccine type and described in a table legend.

5.       The switch from BTV nanoparticle vaccines (Sections 3.1 and 3.2) to an extremely detailed overview of the mNS avian reovirus vaccine system (Section 3.3) is a bit abrupt and confusing. Also, this section contains a lot of superfluous information that is not relevant to its use as a vaccine platform. If a new section is included to describe the nanoparticle and microparticle vaccine approaches (minor comment #1), the background on the mNS avian reovirus vaccine approach can be moved to that section since this platform is used for multiple viruses discussed. Then, in each virus section, the vaccination and outcome can be discussed.

6.       Please define MVA in the text.

7.       There are many serotypes for BTV, AHSV, and EHDV. Is one serotype of each virus more dominant in nature? Are the serotypes localized to geographically distinct regions or are they widespread?

Author Response

Major Comments:

  1. Figure 1 does not support the content of the review. It would be better suited to present the organization of the genera and families as a phylogenetic tree with the viruses and serotypes that are discussed included.

As the reviewer has pointed out, Figure 1 does not add any significant information in the context of vaccines against orbiviruses. In this sense, we have eliminated figure 1 from the text. Regarding the addition of a phylogenetic tree including the three viruses discussed in this work along with their serotypes, we think that this data has been extensively showed across bibliography, with cross-neutralization relationships between serotypes (specially for BTV and AHSV) well characterized. In this sense, the addition of a phylogenetic analysis could be redundant or distracting for the reader.

  1. An additional figure with an illustration of a cross-section of the orbivirus virion would provide a better reference for the readers for the proteins that are expressed to produce the VLPs and CLPs.

The reviewer is right. Despite the composition of VLPs and CLPs is described a few times throughout the manuscript, this work would benefit from including an additional figure illustrating the composition of an orbivirus virion along with the composition of VLPs and CLPs. We have added this illustration in line 457.

  1. Much of the second half of section 2, describing the immune response to nano- and microparticles, is primarily describing a generic immune response. The authors should summarize the more common concepts and include information on the protective immune response required for orbivirus infections.

We agree with the reviewer`s view on this point. The second half of section 2 might be too generic. We have included information related with the protective immune response elicited after orbivirus infection. We believe this information will help to understand the follow-up text on immunogenicity and protection induced by nano and microparticle vaccine candidates against these viral diseases.

Minor Comments:

  1. The in-depth description of nanoparticle and microparticle vaccines in the introduction (lines 68-155) should be reworked into a new section with subsections for each vaccine approach. This will enhance the flow and readability of the review.

As pointed out by the reviewer’s, we have included a new section named “Nano- and microparticle-based vaccine platforms” where we included general information related with the diversity of nano or microplatforms as well as we depict the background of the novel technology based on avian reovirus muNS particles.

  1. In line 211, it is unclear why the authors described immunoglobulin class switching (or isotype switching) as “so-called antibody switching”. Please adjust.

The reviewer is right. We have eliminated the expression “so-called antibody switching” from the text and we have also modified this expression in Figure 1b.

  1. Please include the endemic regions of BTV.

We have include some Information reflecting the distribution of BTV in lines 424-426:

“Traditionally, BTV global distribution is enclosed to regions between approximately 50°N and 35°S. Nonetheless, northwards expansion has occurred during recent years, being identified in all continents except Antartica”

  1. For Table 1 and 2, please include how many doses of the vaccine were administered and elaborate on how protection was assessed, e.g., survival, viral burden, etc. If space is limiting, the expression system can be denoted as a symbol on the vaccine type and described in a table legend.

We totally agree with the reviewer`s comment on this point as it is important to know how many doses of a vaccine candidate were inoculated as well as the parameters that were used to study the protection induced by any vaccine candidate under evaluation. Therefore, we have included an additional column that reflects the number of doses and the dose of protein inoculated, and we have added information regarding the parameters studied to assess protection within the column named “Protection”. We have also followed reviewer`s recommendation on the use of symbols to denote the expression system on the tables.

  1. The switch from BTV nanoparticle vaccines (Sections 3.1 and 3.2) to an extremely detailed overview of the mNS avian reovirus vaccine system (Section 3.3) is a bit abrupt and confusing. Also, this section contains a lot of superfluous information that is not relevant to its use as a vaccine platform. If a new section is included to describe the nanoparticle and microparticle vaccine approaches (minor comment #1), the background on the mNS avian reovirus vaccine approach can be moved to that section since this platform is used for multiple viruses discussed. Then, in each virus section, the vaccination and outcome can be discussed.

We agree with the reviewer and we have incorporated a new section that comprises general data of the different nanoparticle and microparticle platforms approaches along with the background on the mNS avian reovirus vaccine approach. Superfluous information on this issue has been removed.

  1. Please define MVA in the text.

We have included the meaning of the abbreviation MVA (Modified Vaccinia Ankara virus) in line 525.

  1. There are many serotypes for BTV, AHSV, and EHDV. Is one serotype of each virus more dominant in nature? Are the serotypes localized to geographically distinct regions or are they widespread?

BTV, AHSV and EHDV serotypes distribution is highly dinamic. In the case of BTV, the presence of individual serotypes is highly heterogeneous throughout the globe (see Saminathan M. et al. 2020 doi: 10.1080/01652176.2020.1831708 and Maclachlan 2011 doi: 10.1016/j.prevetmed.2011.04.005.). For AHSV, serotype 9 is widespread in endemic areas. Serotypes 1 to 8 occurred in limited areas and serotypes 2, 4, 6, 7 and 8 have co-circulated with AHSV-9 (see Zientara, S., Weyer, C. T., & Lecollinet S., 2015 doi: 10.20506/rst.34.2.2359). The global distribution of EHDV is less well characterized although probably very similar to that of BTV. This virus has been detected in America, Asia, Africa, the Middle East and Oceania. Recently, serotypes 2, 6 and 7 have caused outbreaks in northern and southern Africa, North America, Reunion Island, and the Mediterranean Basin, including Algeria, Israel, Jordan, Morocco, Tunisia and western Turkey (see Noronhan, E. L., 2021 doi: 10.3390/v13112268). Regarding the dominance of a specific serotype in nature, it relies on a variety of factors that comprise the presence and type of host, virus strain, vector population and environmental conditions within a specific region. Recently, it has been described that point mutations can influence vector competence (van Gennip, RGP., et al., 2019 doi: 10.1186/s13071-019-3722-2). In any case, serotype prevalence can vary between affected regions.

Reviewer 2 Report

Review:1776254

Title: Nano- and microparticles-based vaccines against orbiviruses of veterinary importance.

Authors: Jimenez-Cabello et al.

This review seems to have as its main focus the use of virus-like particles as a practical vaccine methodology for immunization of mainly three Reoviridae members of veterinary concern as are blue tongue virus, African horse sickness virus, and epizootic hemorrhagic disease virus.

In general terms, the review is well written—however, there are some misconceptions in the manuscripts.

1)    Thus, the review, according to the title and the introduction, focuses on the role of VLPs and CLPs in triggering immunity BTV, AHSV, and EHDV. VLPs and CLPs are considered nano particles which is consistent with the author's proposed title. Then, the authors discuss broadly ARV C-terminal region muNS ability to form cytosolic inclusions in mammalian cells, yeast, and bacteria, as well as their putative use as a vaccine. However, muNS and variants are not considered as particles but instead as inclusions having even liquid-liquid phase separated like properties. Therefore, they cannot be regarded as particles. In this context, my comments are to change the title of the review to a broader one and to move the paragraphs regarding muNS platforms to the introduction.

2)     Additionally, as currently written, this manuscript is highly biased by the author's research. Please expand the review to other research in muNS platforms or other alternative vaccination methods without emphasizing your own proposed results.

Minor comments
Figure 1 is unnecessary in this manuscript and needs to be removed.

The sentence in lines 34 -35 is a little wild. This is because Reoviridae members effectively infect a broad spectrum of the host, including animals (fish, mammals, birds, and insects), plant kingdom, and fungi kingdom members that need to be included in your sentence.

In paragraph 128-154, it is discussed the advantages of using VLPs in contrast to other vaccination platforms. However, it does not refer to this particular system's disadvantages, like the price and cumbersome quality controls.

In line 285: Change order to …(His 487 and Cys 489)…

Lines 523 to 527. Please revise the sentence. It is hard to read and change the symbol $ by a specific written currency.

Author Response

This review seems to have as its main focus the use of virus-like particles as a practical vaccine methodology for immunization of mainly three Reoviridae members of veterinary concern as are blue tongue virus, African horse sickness virus, and epizootic hemorrhagic disease virus.

In general terms, the review is well written—however, there are some misconceptions in the manuscripts.

  • Thus, the review, according to the title and the introduction, focuses on the role of VLPs and CLPs in triggering immunity BTV, AHSV, and EHDV. VLPs and CLPs are considered nano particles which is consistent with the author's proposed title. Then, the authors discuss broadly ARV C-terminal region muNS ability to form cytosolic inclusions in mammalian cells, yeast, and bacteria, as well as their putative use as a vaccine. However, muNS and variants are not considered as particles but instead as inclusions having even liquid-liquid phase separated like properties. Therefore, they cannot be regarded as particles. In this context, my comments are to change the title of the review to a broader one and to move the paragraphs regarding muNS platforms to the introduction.

We respectfully disagree with the reviewer: to the best of our knowledge there is no evidence that muNS have “liquid-liquid phase” characteristics. We guess that the reviewer extrapolates the results obtained by Geiger et al. (The EMBO Journal (2021) 40:e107711) where the authors show liquid-liquid phase characteristics for Rotavirus viroplasms and inclusions made by NSP2 and NSP5 proteins, but there are many differences with muNS; to cite two of the main ones: muNS is able by itself to form inclusions and it does not bind to RNA at all. But furthermore, the variant used to form micro and nanoparticles, muNS-Mi, is only 1/3 of the total protein, and behaves completely different form the full length muNS. Indeed, one version very close in size and composition to muNS even forms crystals inside cells that are able to diffract (Struct Dyn. 2015 May 22;2(4):041712. doi: 10.1063/1.4921591). But independently of the way that the muNS-Mi/IC-tagged protein associate, once purified from cells, they are in fact and behave like particles. They keep their size and composition after rounds of centrifugation: 1-4 micrometers for those obtained from baculovirus/insect cell system (microparticles, microspheres or MS) and 300-400 nm for those obtained from bacteria (nanoparticles, nanospheres or NS). They do not fuse or dismantle, even after SDS-purification, they are perfectly seen under different microscopic techniques: light microscope, TEM, SEM, AFM. They can also be counted, tracked and immunocharacterized by cytometry. They are even amenable to in vitro PEG-ylation and they suffer the typical Z-potential and apparent diameter changes of any other partciles described in the literature when analyzed by DLS. Thus, we truly believe on their particle nature.

  • Additionally, as currently written, this manuscript is highly biased by the author's research. Please expand the review to other research in muNS platforms or other alternative vaccination methods without emphasizing your own proposed results.

We understand that all muNS platform-related data could be biased by author’s research. Actually, the innovative and promising muNS technology has been only applied as an antigen delivery platform for the orbiviruses included in this review and for Rift Valley Fever virus, and it has been patent by Dr. Martínez-Costas. There is no other research in muNS platform. In any case, we have removed superfluous information on this technology to reduce the bias observed by the reviewer.

Minor comments
Figure 1 is unnecessary in this manuscript and needs to be removed.

As the reviewer has pointed out, Figure 1 does not support the content of the review. In this sense, we have eliminated figure 1 from the text.

The sentence in lines 34 -35 is a little wild. This is because Reoviridae members effectively infect a broad spectrum of the host, including animals (fish, mammals, birds, and insects), plant kingdom, and fungi kingdom members that need to be included in your sentence.

In view of the reviewer`s comment, we have changed lines 34-35 so that we affirm that Reoviridae members can effectively infect fish, mammals, birds, insects, plants and fungi:

The order Reovirales is a heterogeneous non enveloped virus group that comprises a variety of viruses widely distributed in nature through a diversity of hosts, ranging from birds, fish and mammals to insects, fungi and plants, leading to different pathological manifestations”.

In paragraph 128-154, it is discussed the advantages of using VLPs in contrast to other vaccination platforms. However, it does not refer to this particular system's disadvantages, like the price and cumbersome quality controls.

We have added some information reflecting the disadvantages of VLPs in lines 223-226, mainly focused on the downstream processing, as indicated by the reviewer:

“Importantly, VLPs imply some significant technical challenges, e.g. low stability and instability throughout the downstream processing, variable expression of different proteins between expression systems or co-purification of impurities that may affect VLP immunogenicity”

In line 285: Change order to …(His 487 and Cys 489)…

We have changed the order as indicated by the reviewer.

Lines 523 to 527. Please revise the sentence. It is hard to read and change the symbol $ by a specific written currency.

We agree with the reviewer`s comment. We have now revised and changed this sentence to make it more readable:

“To date, there is no evidence of the presence of EHDV in Europe. However, its presence in this continent is feasible as EHDV has been detected in countries of the Mediterranean Basin such as Morocco, Algeria, Tunisia, Israel, Jordan and Turkey and this virus shares epidemiologic similarities with BTV”

We have also substituted the symbol $ for US dollars in lines 427 and 753.

Round 2

Reviewer 1 Report

The authors have adequately addressed my initial comments. 

Minor Comments from revisions:

1. Line 271-272, please add CD8+ T cells since they are activated by MHC-I . "...MHC-I or MHC-II molecules by APCs to trigger the activation of CD8+ T cells or Th CD4+ cells, respectively." 

2. I appreciated the addition of Figure 3. However, the color for VP2 is different between the virion, VLP, and CLP and the legend. 

3. The additional information on immunogenicity and/or protection in table 1 and 2 is a great addition. I understand why the the authors used to superscript for specific disease characterization, but it would be useful to elaborate a bit on what was observed. For example, in the table legend, define that the protection was determined by cincreased survival, dreduced viremia, etc. This way the reader does not have to assume what protection implies. 

Author Response

We have followed the three indications (minor points) suggested by reviewer 1 and the manuscript has been modified accordingly.